# Optimal conversion from Rényi Differential Privacy to $f$-Differential Privacy

**Anneliese Riess** [1 2 3]   **Juan Felipe Gomez** [4]   **Flavio du Pin Calmon** [4]   **Julia A. Schnabel** [1 3 5 6]   **Georgios Kaissis** [7]

## Abstract

We prove the conjecture stated in Appendix F.3 of Zhu et al. (2022): among all conversion rules that map a Rényi Differential Privacy (RDP) profile $\tau \mapsto \rho(\tau)$ to a valid hypothesis-testing trade-off $f$, the rule based on the intersection of single-order RDP privacy regions is optimal. This optimality holds simultaneously for all valid RDP profiles and for all Type I error levels $\alpha$. Concretely, we show that in the space of trade-off functions, the tightest possible bound is $f_{\rho(\cdot)}(\alpha) = \sup_{\tau \geq 0.5} f_{\tau,\rho(\tau)}(\alpha)$: the pointwise maximum of the single-order bounds for each RDP privacy region. Our proof unifies and sharpens the insights of Balle et al. (2019), Asoodeh et al. (2021), and Zhu et al. (2022). Our analysis relies on a precise geometric characterization of the RDP privacy region, leveraging its convexity and the fact that its boundary is determined exclusively by Bernoulli mechanisms. Our results establish that the "intersection-of-RDP-privacy-regions" rule is not only valid, but optimal: no other black-box conversion can uniformly dominate it in the Blackwell sense, marking the fundamental limit of what can be inferred about a mechanism's privacy solely from its RDP guarantees.

## 1. Introduction

The hypothesis testing interpretation of Differential Privacy (DP), often formalized as $f$-DP (Dong et al., 2019), has emerged as a rigorous standard for privacy analysis, grounding privacy guarantees in the operational framework of binary hypothesis testing. By characterizing the trade-off between Type I and Type II errors that an adversary faces when distinguishing between adjacent datasets; $f$-DP provides a complete and geometrically interpretable picture of privacy loss. However, despite the rise of numerical accounting tools, Rényi Differential Privacy (RDP) (Mironov, 2017) remains indispensable for many tasks due to its analytical tractability. For example, some individual privacy accounting techniques (Feldman & Zrnic, 2021) and hyperparameter transfer approaches (Papernot & Steinke, 2021) rely entirely on RDP, making it often the only tractable accounting tool available for advanced algorithms. Beyond composition, RDP is equally central to private selection: the exponential mechanism is most tightly analyzed via zero-Concentrated Differential Privacy (Cesar & Rogers, 2021), which maps directly to an RDP profile. This practical indispensability extends to complex domains such as graph learning (Xiang et al., 2025).

Unlike $f$-DP or Total Variation privacy (Kulynych et al., 2025), which admit direct variational representations involving a single rejection region, the Rényi divergence does not admit a direct hypothesis-testing interpretation for general distributions (Balle et al., 2019). To bridge the gap between the calculable moments of RDP and the interpretable error trade-offs of $f$-DP, one must rely on the "2-cut" reduction (Balle et al., 2019) to analyze what a divergence constraint means in terms of binary hypothesis testing. This necessity drives the construction of the *RDP privacy regions*: the set of all possible error pairs $(\alpha, \beta)$ compatible with a given RDP guarantee.

In this work, we address the problem of optimally converting an RDP profile into $f$-DP. While prior works, such as Asoodeh et al. (2021), have solved the variational problem for a single Rényi order $\tau$, a mechanism typically satisfies a continuum of constraints defined by a profile $\tau \mapsto \rho(\tau)$ (also known as functional RDP (Wang et al., 2018)). We prove the conjecture stated in Appendix F.3 of Zhu et al. (2022): among all conversion rules that map an RDP profile to a valid hypothesis-testing trade-off $f$, the lower boundary of the intersection of $\tau$-RDP privacy regions over all $\tau \in [0.5, \infty)$ is optimal. Any tighter conversion rule would necessarily require information about the mechanism be-

[1]Institute of Machine Learning in Biomedical Imaging, Helmholtz Munich, Neuherberg, Germany [2]School of Computation, Information and Technology, Technical University of Munich, Munich, Germany [3]Munich Center for Machine Learning, Munich, Germany [4]Harvard University, Cambridge, MA, USA [5]Institute for Computational Imaging and AI in Medicine, Technical University of Munich, Munich, Germany [6]School of Biomedical Engineering & Imaging Sciences, King's College London, UK [7]Hasso Plattner Institute for Digital Engineering, University of Potsdam, Potsdam, Germany. Correspondence to: Anneliese Riess <anne.riess@tum.de>.

*Proceedings of the 43rd International Conference on Machine Learning*, Seoul, South Korea. PMLR 306, 2026. Copyright 2026 by the author(s).

yond its RDP profile. We solve the functional optimization problem over this entire trajectory, seeking the tightest possible envelope that holds for *all* mechanisms satisfying a given RDP profile. Our proof unifies and sharpens the insights of Balle et al. (2019), Asoodeh et al. (2021), and Zhu et al. (2022).

Our main contribution is establishing the fundamental limit of black-box privacy conversion, where the conversion relies solely on the RDP profile $\rho(\cdot)$ and is oblivious to all other properties of the underlying privacy mechanism. We prove that the trade-off function derived from the intersection of privacy regions for all Rényi orders is the pointwise optimal one. By constructing witness mechanisms, namely, specific instances of Randomized Response that exactly saturate this bound, we demonstrate that no tighter conversion rule can exist without inspecting other properties of the mechanism. This result elevates the proposed conversion from a technical improvement to a definitive conclusion for RDP-to-DP conversion research: we have reached the theoretical ceiling of what can be inferred solely from RDP parameters.

## 2. Preliminaries

We assume familiarity with the fundamental concepts of differential privacy; however, to ensure this work is self-contained, we review the necessary definitions and notation in this section.

We denote by $\mathbb{D}$ the universe of all possible datasets. Let $(\mathcal{Y}, \Sigma)$ be a measurable space, and let $\mathcal{P}(\mathcal{Y})$ denote the set of all probability measures on $\mathcal{Y}$.

### 2.1. Privacy Definitions via Divergences

A randomized mechanism $\mathcal{M} : \mathbb{D} \to \mathcal{P}(\mathcal{Y})$ maps a dataset $\mathcal{D}$ to a probability distribution $\mathcal{M}(\mathcal{D})$. We define the adjacency relation $\mathcal{D} \sim \mathcal{D}'$ for datasets differing by a single record.

**Definition 2.1** (($\varepsilon, \delta$)-Differential Privacy). A mechanism $\mathcal{M}$ satisfies ($\varepsilon, \delta$)-DP if and only if for all $\mathcal{D} \sim \mathcal{D}'$:

$$E_{e^\varepsilon}(\mathcal{M}(\mathcal{D})\|\mathcal{M}(\mathcal{D}')) \leq \delta, \tag{1}$$

where $E_\gamma(P\|Q) := \int_{\mathcal{Y}} \max(0, p(y) - \gamma q(y))d\mu(y)$ is the hockey-stick divergence, with $p$ and $q$ denoting the densities of $P$ and $Q$ with respect to a dominating measure $\mu$. We use $\gamma = e^\varepsilon$ throughout.

**Definition 2.2** (Rényi Differential Privacy (Mironov, 2017)). A mechanism $\mathcal{M}$ satisfies ($\tau, \rho$)-RDP if and only if for all $\mathcal{D} \sim \mathcal{D}'$:

$$D_\tau(\mathcal{M}(\mathcal{D})\|\mathcal{M}(\mathcal{D}')) \leq \rho, \tag{2}$$

where $D_\tau(P\|Q) := \frac{1}{\tau-1} \log \mathbb{E}_{y \sim Q}[(p(y)/q(y))^\tau]$ is the Rényi divergence of order $\tau \geq 1$, with $\tau = 1$ defined by continuous extension as $D_1(P\|Q) := D_{\mathrm{KL}}(P\|Q)$.

A mechanism typically satisfies a family of RDP guarantees rather than a single one. The function $\rho(\cdot)$ for which $D_\tau(\mathcal{M}(\mathcal{D})\|\mathcal{M}(\mathcal{D}')) \leq \rho(\tau)$ holds for all $\mathcal{D} \sim \mathcal{D}'$ and all $\tau$ is called the *RDP profile* of $\mathcal{M}$.

**Extended Domain.** While the standard definition of RDP focuses on orders $\tau > 1$ (Mironov, 2017), Balle et al. (2019) demonstrated that the constraints corresponding to $\tau \in (0, 1)$ are essential for a tight geometric characterization of the privacy region: omitting them yields strictly suboptimal conversions, including for the Gaussian mechanism (see Figure 3). Because $D_\tau(P\|Q)$ is well-defined for all $\tau \in (0, \infty)$ (assuming $P$ and $Q$ have common support), we treat RDP profiles as functions on the extended domain $[0.5, \infty)$. The symmetry $D_\tau(P\|Q) = \frac{\tau}{1-\tau}D_{1-\tau}(Q\|P)$ for $\tau \in (0, 1)$ (van Erven & Harremoes, 2014) implies that constraints for $\tau \in (0, 0.5)$ are redundant given constraints for $\tau \in [0.5, 1)$; we therefore restrict attention to $\tau \in [0.5, \infty)$ throughout. A formal justification is given in Appendix A.

Rényi divergence is non-decreasing in $\tau$, therefore $\rho(\cdot)$ is non-decreasing as well. Any profile $\rho(\cdot)$ is automatically *valid*, i.e., the class of compatible mechanisms is non-empty, since the perfectly-private mechanism, with divergence and profile identically zero, belongs to every such class. However, for completeness, we note that a profile is *achievable*, i.e., equal to the exact RDP of some mechanism rather than merely an upper bound, if and only if $(\tau - 1)\rho(\tau)$ is convex in $\tau$, the cumulant generating function characterization of the privacy loss random variable (Wang et al., 2018; Balle et al., 2019). Our results hold for all valid profiles.

### 2.2. Hypothesis Testing

The operational interpretation of DP is best understood through binary hypothesis testing. Consider the task of distinguishing between two adjacent datasets, $\mathcal{D}$ and $\mathcal{D}'$, based on the output $Y \in \mathcal{Y}$ of a mechanism $\mathcal{M}$. This task corresponds to testing the null hypothesis $H_0 : Y \sim P$ against the alternative $H_1 : Y \sim Q$, where $P = \mathcal{M}(\mathcal{D})$ and $Q = \mathcal{M}(\mathcal{D}')$. A decision rule is defined by a rejection region $S \subseteq \mathcal{Y}$ (where we reject $H_0$ if $Y \in S$). This induces two types of errors:

$$\text{Type I Error:} \quad P(S) = \mathbb{P}_{Y \sim P}(Y \in S),$$
$$\text{Type II Error:} \quad Q(S^c) = \mathbb{P}_{Y \sim Q}(Y \notin S).$$

The difficulty of distinguishing $P$ from $Q$ is fully characterized by the trade-off function $T(P, Q)$, which maps a Type I error level $\alpha \in [0, 1]$ to the minimal possible Type II error:

$$T(P, Q)(\alpha) := \inf_{S \in \Sigma}\{Q(S^c) : P(S) \leq \alpha\}. \tag{3}$$

Let $f$ be a convex, non-increasing function on the unit square. Then, $f$-DP (Dong et al., 2019) ensures that for

*any* adjacent datasets $\mathcal{D} \sim \mathcal{D}'$, with $P = \mathcal{M}(\mathcal{D})$ and $Q = \mathcal{M}(\mathcal{D}')$:

$$T(P,Q)(\alpha) \geq f(\alpha), \quad \forall \alpha \in [0,1].$$

This effectively limits the power of any adversary to identify the source dataset.

Lastly, we say that two DP mechanisms are Blackwell equivalent if and only if their trade-off functions coincide everywhere (Kaissis et al., 2025). Moreover, we say a mechanism with trade-off function $f$ dominates another with trade-off function $g$ in the Blackwell sense if and only if $f(\alpha) \geq g(\alpha)$ for all $\alpha \in [0,1]$.

### 2.3. Conversion Rules and RDP Classes

In many settings, such as private deep learning, the specific mechanism $\mathcal{M}$ and datasets $\mathcal{D}, \mathcal{D}'$ are unknown or effectively black-box; only the privacy accountant's output, i.e. the RDP profile $\rho(\tau)$, is available. Therefore, since we cannot inspect the mechanism directly, finding a suitable $f$-DP guarantee requires characterizing the trade-off function over the entire class of distributions permissible under $\rho$.

Let $\mathcal{S}_\rho$ be the set of all pairs of probability distributions $(P,Q)$ that satisfy the RDP constraints:

$$\mathcal{S}_\rho := \left\{ (P,Q) \in \mathcal{P}(\mathcal{Y})^2 \; \middle| \; \begin{array}{l} D_\tau(P\|Q) \leq \rho(\tau) \\ D_\tau(Q\|P) \leq \rho(\tau) \end{array} \forall \tau \geq 0.5 \right\}.$$

**Definition 2.3** (Admissible Conversion Rule). Let $\Pi$ be the space of valid RDP profiles and $\mathcal{T}$ the space of trade-off functions. A transformation $C : \Pi \to \mathcal{T}$ is an *admissible conversion rule* if for any profile $\rho \in \Pi$, it lower-bounds the trade-off of every pair in $\mathcal{S}_\rho$:

$$C(\rho)(\alpha) \leq \inf_{(P,Q) \in \mathcal{S}_\rho} T(P,Q)(\alpha), \quad \forall \alpha \in [0,1]. \quad (4)$$

This definition ensures that $C(\rho)$ is a valid lower bound on the hypothesis testing difficulty. In the simplified case where we possess only a single point-wise guarantee $(\tau^*, \epsilon^*) \in [0.5, \infty) \times \mathbb{R}_{\geq 0}$ rather than a full functional profile, we define the conversion rule analogously by considering the profile $\rho$ where $\rho(\tau^*) = \epsilon^*$ and $\rho(\tau) = \infty$ for all $\tau \neq \tau^*$.

### 2.4. Privacy Region

We define the *privacy region* of a mechanism as the set of all attainable error pairs $(\alpha, \beta)$ for the binary hypothesis testing problem. Wasserman & Zhou (2009) demonstrated that a mechanism is $(\varepsilon, \delta)$-DP if and only if all attainable error pairs lie within the region $R_{\mathrm{DP}}(\varepsilon, \delta)$:

$$R_{\mathrm{DP}}(\varepsilon, \delta) = \left\{ (\alpha, \beta) \in [0,1]^2 \; \middle| \; \begin{array}{l} 1 - \alpha \leq e^\varepsilon \beta + \delta \\ 1 - \beta \leq e^\varepsilon \alpha + \delta \end{array} \right\}. \quad (5)$$

The lower boundary of this region is the *tightest lower-bounding* piecewise linear trade-off function $f_{\varepsilon,\delta} : [0,1] \to [0,1]$, given by:

$$f_{\varepsilon,\delta}(\alpha) = \max \left\{ 0, 1 - \delta - e^\varepsilon \alpha, e^{-\varepsilon}(1 - \delta - \alpha) \right\}.$$

The set $R_{\mathrm{DP}}(\varepsilon, \delta)$ represents the collection of permissible error rates. For a mechanism to satisfy $(\varepsilon, \delta)$-DP, no adversary can construct a test $S$ with error rates $(\alpha, \beta)$ falling outside this region (i.e., closer to the origin $(0,0)$ than the boundary allows).

Using the $(\varepsilon, \delta)$-DP definition via the hockey-stick divergence (see Definition 2.1), we can extend the definition of privacy regions to classes of distributions. Consider the set of all distribution pairs consistent with the privacy parameters:

$$\mathcal{S}_{\varepsilon,\delta} := \left\{ (P,Q) \in \mathcal{P}(\mathcal{Y})^2 \; \middle| \; \begin{array}{l} E_{e^\varepsilon}(P\|Q) \leq \delta \\ E_{e^\varepsilon}(Q\|P) \leq \delta \end{array} \right\}.$$

The privacy region $R_{\mathrm{DP}}(\varepsilon, \delta)$ is precisely the set of all error pairs $(\alpha, \beta)$ attainable by any binary hypothesis test trying to distinguish between any pair $P$ and $Q$ in $\mathcal{S}_{\varepsilon,\delta}$.

#### 2.4.1. THE RDP PRIVACY REGION AND 2-CUTS

Privacy regions can also be constructed using divergences such as the Rényi divergence (Balle et al., 2019). However, this requires an additional step employing the concept of $k$-cuts (Balle et al., 2019), specifically the 2-cut, which relates high-dimensional distributions to binary hypothesis testing. Unlike the Total Variation distance, the Rényi divergence lacks a direct variational representation involving a single rejection region, and therefore cannot be translated into error trade-offs without first projecting onto binary outcomes. Intuitively, the 2-cut reduction projects the distinguishability (measured by some divergence) of complex high-dimensional distributions onto this reduced space, where privacy loss can be expressed directly in terms of Type I and Type II errors.

Consider a mechanism satisfying $(\tau, \rho)$-RDP. For any decision rule defined by rejection region $S \subseteq \mathcal{Y}$, we can define a randomized binary test with Type I error $\alpha = P(S)$ and Type II error $\beta = Q(S^c)$. These errors induce two Bernoulli distributions:

$$\mathcal{B}_P \sim \mathrm{Bern}(\alpha) \quad \text{and} \quad \mathcal{B}_Q \sim \mathrm{Bern}(1 - \beta). \quad (6)$$

The data processing inequality (DPI) guarantees that post-processing (in this case, mapping the mechanism's output to a binary decision) cannot increase the divergence. The 2-cut of the Rényi divergence, denoted $\bar{D}_\tau^2(P\|Q)$, can be interpreted as the worst-case divergence achievable by any such binary reduction:

$$\bar{D}_\tau^2(P\|Q) := \sup_{S \subseteq \mathcal{Y}} D_\tau(\mathcal{B}_P\|\mathcal{B}_Q).$$

By the DPI, this quantity is upper-bounded by the divergence of the original distributions, which is in turn bounded by the privacy budget $\rho$:

$$D_\tau(\mathcal{B}_P \| \mathcal{B}_Q) \leq \bar{D}_\tau^2(P\|Q) \leq D_\tau(P\|Q) \leq \rho. \quad (7)$$

Since (7) holds for every test $S$, the 2-cut provides a necessary condition on all attainable error pairs: any $(\alpha, \beta)$ achievable by a $(\tau, \rho)$-RDP mechanism must satisfy $D_\tau(\mathcal{B}_P \| \mathcal{B}_Q) \leq \rho$. We define the $\tau$-order RDP privacy region, denoted by $R_{D_\tau}(\rho)$, as the set of all error pairs $(\alpha, \beta) \in [0,1]^2$ attainable by some mechanism satisfying $(\tau, \rho)$-RDP and some binary test. Formally, $(\alpha, \beta) \in R_{D_\tau}(\rho)$ if and only if there exist distributions $P, Q$ and a test $S \subseteq \mathcal{Y}$ such that: $D_\tau(P\|Q) \leq \rho$ and $\quad D_\tau(Q\|P) \leq \rho$, with $P(S) \leq \alpha$ and $Q(S^c) \leq \beta$.

**Proposition 2.4** (Bernoulli Characterization of the RDP Privacy Region). *The $\tau$-order RDP privacy region admits the explicit characterization:*

$$R_{D_\tau}(\rho) =$$
$$\left\{ (\alpha, \beta) \in [0,1]^2 \,\middle|\, \begin{array}{l} D_\tau(\mathrm{Bern}(\alpha)\|\mathrm{Bern}(1-\beta)) \leq \rho \\ D_\tau(\mathrm{Bern}(1-\beta)\|\mathrm{Bern}(\alpha)) \leq \rho \end{array} \right\}. \quad (8)$$

*Proof.* We prove equality by double inclusion.

($\subseteq$) Let $(\alpha, \beta) \in R_{D_\tau}(\rho)$. Then by definition there exist distributions $P, Q$ with $D_\tau(P\|Q) \leq \rho$, $D_\tau(Q\|P) \leq \rho$, and a test $S$ with $P(S) = \alpha$ and $Q(S^c) = \beta$. The test $S$ induces Bernoulli distributions $\mathcal{B}_P \sim \mathrm{Bern}(\alpha)$ and $\mathcal{B}_Q \sim \mathrm{Bern}(1-\beta)$. By (7): $D_\tau(\mathrm{Bern}(\alpha)\|\mathrm{Bern}(1-\beta)) \leq D_\tau(P\|Q) \leq \rho$, and analogously $D_\tau(\mathrm{Bern}(1-\beta)\|\mathrm{Bern}(\alpha)) \leq \rho$.

($\supseteq$) Suppose $(\alpha, \beta)$ satisfies both Bernoulli constraints. Set $P = \mathrm{Bern}(\alpha)$ and $Q = \mathrm{Bern}(1-\beta)$. By assumption, $D_\tau(P\|Q) \leq \rho$ and $D_\tau(Q\|P) \leq \rho$, so $(P, Q)$ is a valid $(\tau, \rho)$-RDP mechanism. The identity test $S = \{1\}$, which rejects $H_0$ upon observing output 1, achieves $P(S) = \alpha$ and $Q(S^c) = \beta$ exactly, so $(\alpha, \beta) \in R_{D_\tau}(\rho)$. $\square$

Depending on the value of $\tau$, the constraints in (8) take the following explicit forms derived from (7):

**Case 1** ($\tau > 1$): For $\tau > 1$, the region is the set of $(\alpha, \beta)$ satisfying:

$$\alpha^\tau (1-\beta)^{1-\tau} + (1-\alpha)^\tau \beta^{1-\tau} \leq e^{(\tau-1)\rho}$$
$$(1-\beta)^\tau \alpha^{1-\tau} + \beta^\tau (1-\alpha)^{1-\tau} \leq e^{(\tau-1)\rho}. \quad (9)$$

**Case 2** ($\tau = 1$): As $\tau \to 1$, the RDP constraint converges to the KL-divergence:

$$\alpha \log \frac{\alpha}{1-\beta} + (1-\alpha) \log \frac{1-\alpha}{\beta} \leq \rho$$
$$(1-\beta) \log \frac{1-\beta}{\alpha} + \beta \log \frac{\beta}{1-\alpha} \leq \rho. \quad (10)$$

**Case 3** ($0 < \tau < 1$): For $\tau < 1$, the definition of Rényi divergence involves a factor of $\frac{1}{\tau-1} < 0$, which reverses the inequality direction:

$$\alpha^\tau (1-\beta)^{1-\tau} + (1-\alpha)^\tau \beta^{1-\tau} \geq e^{(\tau-1)\rho}$$
$$(1-\beta)^\tau \alpha^{1-\tau} + \beta^\tau (1-\alpha)^{1-\tau} \geq e^{(\tau-1)\rho}. \quad (11)$$

The constraints (9)–(11) have already been derived in Zhu et al. (2022); however, for the self-containedness of this manuscript, we have included their derivation in Appendix A. These inequalities define the $\tau$-order RDP privacy region $R_{D_\tau}(\rho)$ for $\tau > 1$, $\tau = 1$, and $0 < \tau < 1$, respectively. Note that the $\tau$-order RDP privacy regions for $\tau' \in [0.5, 1)$, are sufficient to characterize the $\tau$-order RDP privacy regions for all $0 < \tau < 1$ (see Apendix A).

### 2.4.2. PROPERTIES OF THE RDP PRIVACY REGION

**Proposition 2.5** (Convexity and Symmetry of the RDP Privacy Region). *For any $\tau \in [0.5, \infty)$ and $\rho \geq 0$, the RDP privacy region $R_{D_\tau}(\rho)$ is a convex set and is symmetric about $\alpha = \beta$.*

*Proof.* It is an established property that the Rényi divergence $D_\tau(P\|Q)$ is jointly quasi-convex in the pair of distributions $(P, Q)$ for all $\tau \in (0, \infty)$ (van Erven & Harremoes, 2014). Consequently, its sublevel sets are convex in the space of probability distributions.

The privacy region $R_{D_\tau}(\rho)$ is defined as the intersection of two sets:

$$S_1 = \{(\alpha, \beta) : D_\tau(\mathrm{Bern}(\alpha)\|\mathrm{Bern}(1-\beta)) \leq \rho\},$$
$$S_2 = \{(\alpha, \beta) : D_\tau(\mathrm{Bern}(1-\beta)\|\mathrm{Bern}(\alpha)) \leq \rho\}.$$

Consider the mapping $M : [0,1]^2 \to \mathcal{P}(\{0,1\})^2$ defined by $M(\alpha, \beta) = (\mathrm{Bern}(\alpha), \mathrm{Bern}(1-\beta))$. Identifying the probability measures with vectors in $\mathbb{R}^2$, we have $\mathrm{Bern}(\alpha) = [1-\alpha, \alpha]^\top$ and $\mathrm{Bern}(1-\beta) = [\beta, 1-\beta]^\top$. The mapping $M$ is affine with respect to the parameters $\alpha$ and $\beta$.

Since the mapping from parameters to distributions is affine, the convexity of the sublevel sets in distribution space is preserved when "pulled back" to the parameter space. Thus, both $S_1$ and $S_2$ are convex sets.

Finally, the privacy region $R_{D_\tau}(\rho) = S_1 \cap S_2$ is the intersection of two convex sets and is therefore convex.

To show symmetry, note that if $(\alpha, \beta) \in R_{D_\tau}(\rho)$, then by definition $D_\tau(\mathrm{Bern}(\alpha)\|\mathrm{Bern}(1-\beta)) \leq \rho$ and $D_\tau(\mathrm{Bern}(1-\beta)\|\mathrm{Bern}(\alpha)) \leq \rho$. Swapping $\alpha$ and $\beta$ in these inequalities yields:

$$D_\tau(\mathrm{Bern}(\beta)\|\mathrm{Bern}(1-\alpha)) \leq \rho$$
$$\text{and} \quad D_\tau(\mathrm{Bern}(1-\alpha)\|\mathrm{Bern}(\beta)) \leq \rho.$$

Since $D_\tau(\text{Bern}(x)\|\text{Bern}(y)) = D_\tau(\text{Bern}(1-x)\|\text{Bern}(1-y))$, the set of constraints is invariant under the transformation $(\alpha, \beta) \mapsto (\beta, \alpha)$.

$\square$

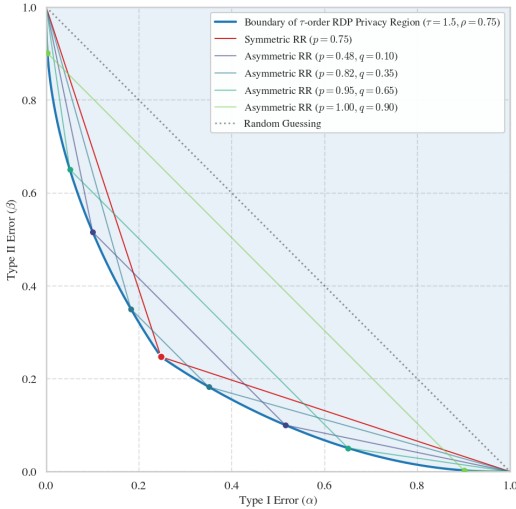

*Figure 1.* Visualization of the $\tau$-order RDP privacy region $R_{D_\tau}(\rho)$ for $\tau = 1.5$ and $\rho = 0.75$. The bold blue line depicts the lower boundary $f_{\tau,\rho}(\alpha)$, and the shaded region corresponds to $R_{D_\tau}(\rho)$. The piecewise linear trade-off functions correspond to specific Randomized Response mechanisms satisfying the $(1.5, 0.75)$-RDP constraint: the red line denotes the symmetric RR mechanism ($p = 0.75$) and the remaining lines denote the asymmetric RR mechanisms with varying parameters $p$ and $q$. For every point on the lower boundary, there exists a Randomized Response mechanism whose trade-off function touches the boundary at exactly that point, establishing that the boundary cannot be tightened without excluding valid mechanisms (Proposition 3.1).

Note that the lower boundary of the RDP privacy region given by $f_{\tau,\rho}(\alpha) = \inf\{\beta : (\alpha, \beta) \in R_{D_\tau}(\rho)\}$ defines a well-behaved trade-off function for all $\alpha \in [0, 1]$:

1. **Convexity:** $f_{\tau,\rho}(\alpha)$ is a convex function. This follows directly from the convexity of the region $R_{D_\tau}(\rho)$ (the lower boundary of a convex set defined on a connected domain is convex).

2. **Monotonicity:** $f_{\tau,\rho}(\alpha)$ is non-increasing. Increasing the allowable Type I error $\alpha$ relaxes the constraints on the test, allowing for a strictly lower (or equal) Type II error $\beta$.

3. **Symmetry:** The RDP region is symmetric about $\alpha = \beta$. Consequently, $(\alpha, \beta))$ is in the boundary if and only if $(\beta, \alpha)$ is on the boundary implying that the trade-off function $f_{\tau,\rho}$ is symmetric.

4. **Feasibility of Random Guessing:** $f_{\tau,\rho}(\alpha) \leq 1 - \alpha$. The trivial "random guessing" test (rejecting $H_0$ with probability $\alpha$ independent of the data) yields $\beta = 1 - \alpha$. This corresponds to the case where the induced binary distributions are identical ($D_\tau = 0 \leq \rho$). Thus, the diagonal line $\beta = 1 - \alpha$ lies strictly inside the region, and the lower boundary must lie below it.

## 3. Optimality of the Single-Order RDP Conversion

We now establish that the boundary of the single-order RDP privacy region is not merely a valid bound, but the *optimal* one. No alternative conversion rule can extract a tighter trade-off function from a single $(\tau, \rho)$-RDP guarantee without excluding valid mechanisms.

**Proposition 3.1** (Optimality of the RDP Boundary). *Let $f_{\tau,\rho}$ be the trade-off function defined by the lower boundary of the RDP privacy region $R_{D_\tau}(\rho)$. For any admissible conversion rule $C$ mapping a single RDP guarantee $(\tau, \rho)$ to a lower bound on trade-off functions, $f_{\tau,\rho}$ is the tightest possible bound. That is, for any other valid lower bound $f'$, $f'(x) \leq f_{\tau,\rho}(x)$ for all $x \in [0, 1]$.*

*Proof.* The proof proceeds in three steps: (1) Identifying valid Bernoulli mechanisms that achieve every point on the boundary; (2) characterizing their trade-off functions; and (3) proving optimality by contradiction.

**Step 1: Achievability by Bernoulli Mechanisms.** By the Bernoulli characterization of $R_{D_\tau}(\rho)$ (see Proposition 2.4), every point $(\alpha^*, \beta^*)$ on the lower boundary satisfies $D_\tau(\text{Bern}(\alpha^*)\|\text{Bern}(1-\beta^*)) \leq \rho$ and $D_\tau(\text{Bern}(1-\beta^*)\|\text{Bern}(\alpha^*)) \leq \rho$. Setting $P = \text{Bern}(\alpha^*)$ and $Q = \text{Bern}(1-\beta^*)$ therefore defines a valid $(\tau, \rho)$-RDP mechanism, which is an instance of Randomized Response (see Appendix B). The test $S = \{1\}$, which rejects $H_0$ upon observing output 1, achieves $P(S) = \alpha^*$ and $Q(S^c) = \beta^*$ exactly.

**Step 2: Validity of the Trade-off Functions.** For any Randomized Response mechanism with output distributions $P = \text{Bern}(\alpha^*)$ and $Q = \text{Bern}(1-\beta^*)$ corresponding to a point $(\alpha^*, \beta^*)$ on the boundary, the full trade-off function is piecewise linear, constructed by combining the two one-sided tests. The test $S = \{1\}$ distinguishing $P$ from $Q$ achieves $(\alpha^*, \beta^*)$, while the reverse test distinguishing $Q$ from $P$ achieves the symmetric point $(\beta^*, \alpha^*)$. The full trade-off function therefore consists of the line segments connecting $(0, 1) \to (\alpha^*, \beta^*) \to (\beta^*, \alpha^*) \to (1, 0)$, with intermediate points attainable via randomization between adjacent tests. In the symmetric case $\alpha^* = \beta^*$, the middle segment collapses to a point and the trade-off reduces to the two segments $(0, 1) \to (\alpha^*, \alpha^*) \to (1, 0)$. In both cases, since $R_{D_\tau}(\rho)$ is convex and symmetric and contains the

vertices $(0, 1)$ and $(1, 0)$, the entire piecewise linear trade-off lies within the region, confirming it is pointwise greater than or equal to the boundary $f_{\tau,\rho}$.

**Step 3: Contradiction.** Suppose, for the sake of contradiction, that there exists an admissible conversion rule yielding a trade-off function $f_{\text{new}}$ that strictly improves upon the boundary. That is, there exists some $x_0 \in [0, 1]$ such that $f_{\text{new}}(x_0) > f_{\tau,\rho}(x_0)$.

From Step 1, we know there exists a valid Randomized Response mechanism $\mathcal{M}_{\text{RR}}$ whose optimal identity test yields exactly the error pair $(x_0, f_{\tau,\rho}(x_0))$. The true trade-off value of $\mathcal{M}_{\text{RR}}$ at Type I error $x_0$ is exactly $f_{\tau,\rho}(x_0)$. However, the conversion rule asserts that *any* mechanism satisfying $(\tau, \rho)$-RDP must have a trade-off at least $f_{\text{new}}(x_0)$. Since $f_{\tau,\rho}(x_0) < f_{\text{new}}(x_0)$, the mechanism $\mathcal{M}_{\text{RR}}$ violates the bound provided by the conversion rule.

This contradicts the assumption that the conversion rule is admissible for the class of $(\tau, \rho)$-RDP mechanisms. Therefore, $f_{\tau,\rho}$ is the tightest possible bound. $\square$

# 4. Validity: Intersection of Regions as a Universal Bound

We now consider the constraint imposed by the entire RDP profile $\rho(\tau)$.

**Lemma 4.1** (Projection of an Intersection). *Let $\{G_\tau\}_\tau$ be a family of closed convex subsets of the unit square whose lower boundaries are given by functions $y = h_\tau(x)$. For fixed $x$, the minimal $y$ such that $(x, y) \in \bigcap_\tau G_\tau$ is given by:*

$$y_{\min}(x) = \sup_\tau h_\tau(x). \tag{12}$$

*Proof.* Fix $x \in [0, 1]$. The feasible set of $y$ values for the intersection is defined as:

$$Y = \bigcap_\tau \{y \in [0, 1] : (x, y) \in G_\tau\}. \tag{13}$$

Since each set $G_\tau$ is defined by a lower boundary $h_\tau$, the condition $(x, y) \in G_\tau$ holds if and only if $y \geq h_\tau(x)$. Consequently, for each $\tau$, the feasible set is the interval $[h_\tau(x), 1]$. The intersection of such intervals is given by:

$$\bigcap_\tau [h_\tau(x), 1] = \left[\sup_\tau h_\tau(x), \, 1\right]. \tag{14}$$

Therefore, the minimal $y$ in the intersection is exactly $\sup_\tau h_\tau(x)$. $\square$

Applying this to the privacy context where $G_\tau = R_{D_\tau}(\rho(\tau))$, this lemma confirms that the optimal trade-off function $f_\rho(\alpha)$ is the pointwise supremum of the single-order lower bounds:

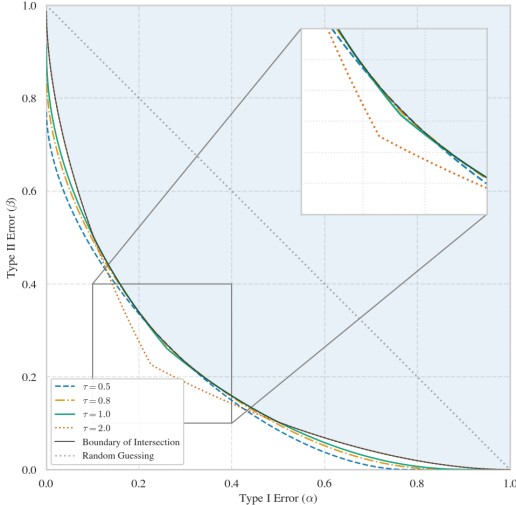

*Figure 2.* Exemplary construction of the joint RDP privacy region $\mathcal{R}_{\text{joint}}$ using a subset of orders $\tau \in \{0.5, 0.8, 1.0, 2.0\}$ for the profile of the Gaussian mechanism with unit sensitivity and noise scale $\sigma = 1$, $\rho(\tau) = \frac{\tau}{2\sigma^2}$. The colored lines depict the lower boundaries of the single-order regions $R_{D_\tau}$. The solid black curve represents the boundary of the intersection, which corresponds to the pointwise maximum (supremum) of the individual single-order boundaries. The inset zooms in on the "tangent" behavior, showing how different orders become active (provide the tightest bound) at different error regimes.

**Corollary 4.2** (Intersection Bound). *If a mechanism satisfies the entire RDP profile $D_\tau \leq \rho(\tau)$ for all $\tau \geq 0.5$, then every attainable error pair $(\alpha, \beta)$ lies in the intersection region:*

$$\mathcal{R}_{\text{joint}} = \bigcap_{\tau \geq 0.5} R_{D_\tau}(\rho(\tau)). \tag{15}$$

*Consequently, the upper envelope defined by $f_{\rho(\cdot)}(\alpha) := \sup_{\tau \geq 0.5} f_{\tau,\rho(\tau)}(\alpha)$ constitutes a valid trade-off function for any mechanism satisfying the profile.*

*Proof.* First, by Theorem 18 of Balle et al. (2019), the assumption $D_\tau \leq \rho(\tau)$ implies that the mechanism's error region is contained in $R_{D_\tau}(\rho(\tau))$ for all $\tau$. Since this inclusion holds for all $\tau \geq 0.5$, the attainable region must lie in the intersection $\bigcap_\tau R_{D_\tau}(\rho(\tau))$.

For the first claim (trade-off function), we apply Lemma 4.1. Since the attainable $(\alpha, \beta)$ lies in the intersection, for any fixed Type I error $\alpha$, the minimal attainable Type II error must be at least the minimal $\beta$ of the intersection boundary. Lemma 4.1 establishes that this boundary is the pointwise supremum of the individual boundaries $f_{\tau,\rho(\tau)}(\alpha)$.

We further confirm that this supremum constitutes a valid trade-off function. It is a standard result in convex analysis that the pointwise supremum of a family of convex functions is convex. Geometrically, since each region $R_{D_\tau}$ is closed and convex, their intersection $\mathcal{R}_{\text{joint}}$ is necessarily a closed,

convex set. Thus, its lower boundary $f_\rho$ is guaranteed to be a valid, convex trade-off curve.

For the second claim (privacy profile), we invoke the geometric containment. For any fixed $\varepsilon \geq 0$, the constraint implies that the mechanism's $\delta(\varepsilon)$ must be consistent with the region $R_{D_\tau}(\rho(\tau))$. Since this inequality must hold simultaneously for every valid $\tau$, the tightest possible bound is the infimum over all $\tau$. $\qquad\square$

*Remark* 4.3 (Geometric Interpretation: The Tangent Constraint). The boundary of the global RDP privacy region $\mathcal{R}_{\text{joint}}$ is formed by the intersection of the infinite family of regions $\{R_{D_\tau}\}_{\tau \geq 0.5}$. Consequently, the final boundary constitutes the upper envelope of the individual boundaries defined by $f_{\tau,\rho(\tau)}$.

For any specific error profile (a point $(\alpha, \beta)$ on the final boundary), there exists a specific optimal order $\tau^*$ such that the boundary of the region $R_{D_{\tau^*}}$ is tangent to the envelope at that point. Geometrically, this implies that the curve defined by the $\tau^*$-constraint and the final envelope curve touch at this point and share the exact same slope (derivative), without crossing. Mathematically, this signifies that while all constraints must be satisfied, the constraint corresponding to $\tau^*$ is the "active" one that locally determines the shape and gradient of the privacy boundary.

With the validity of the intersection bound established, we now have all the necessary components to prove our main result.

**Theorem 4.4** (Universal Optimality of the Intersection-Based Conversion). *For every admissible conversion rule $C$ mapping RDP profiles $\rho$ to lower bounds on trade-off functions, and for every valid profile $\rho$, it holds that:*

$$C(\rho)(\alpha) \leq f_{\rho(\cdot)}(\alpha) \quad \text{for all } \alpha \in [0,1]. \qquad (16)$$

*Equivalently, the intersection-based trade-off $f_{\rho(\cdot)}$ is the pointwise tightest possible black-box conversion from RDP to $f$-DP.*

*Proof.* We prove this by contradiction. Suppose there exists an admissible conversion rule $C$, a profile $\rho$, and a Type I error $\alpha_0 \in [0,1]$ such that the rule yields a strictly tighter bound than the intersection limit:

$$C(\rho)(\alpha_0) > f_{\rho(\cdot)}(\alpha_0). \qquad (17)$$

Let $\beta^* = f_{\rho(\cdot)}(\alpha_0)$. The point $P^* = (\alpha_0, \beta^*)$ lies exactly on the lower boundary of the intersection region $\mathcal{R}_{\text{joint}} = \bigcap_{\tau \geq 0.5} R_{D_\tau}(\rho(\tau))$.

**Step 1: Constructing the Witness Mechanism.** We invoke the characterization of the boundary (see Step 1 of the proof of Proposition 3.1). For the point $P^* = (\alpha_0, \beta^*)$ on the boundary, we can construct a specific binary mechanism

$\mathcal{M}^*$ (an instance of Randomized Response) whose optimal hypothesis test yields the error pair exactly $(\alpha, \beta) = (\alpha_0, \beta^*)$.

**Step 2: Verifying Validity.** We must establish that $\mathcal{M}^*$ is a valid mechanism for the entire profile $\rho$, i.e., it satisfies $D_\tau(\mathcal{M}^*) \leq \rho(\tau)$ for all $\tau \geq 0.5$.

By construction, the point $P^*$ lies within the global intersection $\mathcal{R}_{\text{joint}}$, which implies $P^* \in R_{D_\tau}(\rho(\tau))$ for every individual $\tau$. Crucially, because $\mathcal{M}^*$ is a *binary* mechanism (mapping to $\{0, 1\}$), the DPI for the 2-cut reduction holds with equality. That is, the Rényi divergence of the mechanism is exactly the Rényi divergence of the induced binary distributions defined by its error profile $P^*$:

$$D_\tau(\mathcal{M}^*(\mathcal{D}) \| \mathcal{M}^*(\mathcal{D}')) = D_\tau(\text{Bern}(\alpha_0) \| \text{Bern}(1 - \beta^*)).$$

Since $P^*$ is contained in every region $R_{D_\tau}(\rho(\tau))$, it satisfies the binary divergence constraints for all $\tau$. Therefore, $D_\tau(\mathcal{M}^*) \leq \rho(\tau)$ holds for all $\tau \geq 0.5$, proving that $\mathcal{M}^*$ is a valid instance of the class defined by the profile $\rho$.

**Step 3: Contradiction.** Since $C$ is defined as an admissible conversion rule, it must provide a valid lower bound for the trade-off function of *any* mechanism satisfying the profile $\rho$, including our witness $\mathcal{M}^*$. Evaluating the true trade-off of $\mathcal{M}^*$ at $\alpha_0$, we have $\beta_{\mathcal{M}^*}(\alpha_0) = \beta^*$. Admissibility requires:

$$C(\rho)(\alpha_0) \leq \beta_{\mathcal{M}^*}(\alpha_0) = \beta^*. \qquad (18)$$

However, our initial hypothesis assumed $C(\rho)(\alpha_0) > \beta^*$. This is a contradiction.

Thus, no such strictly tighter bound exists: $f_{\rho(\cdot)}$ is universally optimal and Blackwell dominates any other trade-off function obtainable by black-box conversion from an RDP profile. $\qquad\square$

*Remark* 4.5 (Optimal Conversion for zero-Concentrated Differential Privacy). Our main theorem directly implies an optimal black-box conversion from zero-Concentrated Differential Privacy (zCDP) (Cesar & Rogers, 2021) to $f$-DP. A mechanism satisfying $\gamma$-zCDP admits the parametric RDP profile $\rho(\tau) = \tau\gamma$ for all $\tau \geq 1$ (Cesar & Rogers, 2021). We remark that zCDP is precisely the accounting tool of choice for private selection pipelines combining Gaussian noise and the exponential mechanism (Cesar & Rogers, 2021).

### 4.1. Exactness of the RDP Profile for Randomized Response

As an interesting auxiliary finding, we show that the entire trade-off curve of the Symmetric Randomized Response mechanism is *exactly* recovered by our conversion.

**Proposition 4.6** (Exact Recovery of Randomized Response). *Let $\mathcal{M}_{RR}$ be the Symmetric Randomized Response mech-*

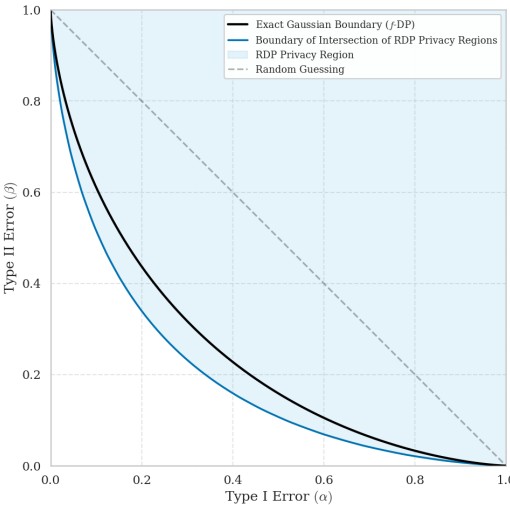

*Figure 3.* Optimal RDP-to-$f$-DP conversion for the Gaussian mechanism. The blue shaded area represents $\mathcal{R}_{\text{joint}}$ for the profile of a Gaussian mechanism with unit sensitivity and noise scale $\sigma = 1$, defined by $\rho(\tau) = \frac{\tau}{2\sigma^2}$. The blue solid line indicates the optimal black-box trade-off function obtained by the intersection of these regions across all orders $\tau \in [0.5, \infty)$. The black solid line indicates the true trade-off for the Gaussian mechanism $f(\alpha) = \Phi(\Phi^{-1}(1 - \alpha) - 1/\sigma)$. Although this conversion is optimal for any mechanism described solely by its RDP profile, it is not exact for all such mechanisms.

*anism with parameter $p > 0.5$, which satisfies pure $\varepsilon$-differential privacy with $\varepsilon = \ln \frac{p}{1-p}$. Let $\rho(\tau)$ be its exact RDP profile for all $\tau \geq 0.5$.*

*The joint RDP privacy region $\mathcal{R}_{joint} = \bigcap_{\tau \geq 0.5} R_{D_\tau}(\rho(\tau))$ coincides exactly with the true attainable privacy region of $\mathcal{M}_{RR}$.*

*Proof.* The proof relies on the behavior of the Rényi divergence as $\tau \to \infty$.

**1. The True Region:** The mechanism $\mathcal{M}_{RR}$ satisfies pure $(\varepsilon, 0)$-differential privacy. Its attainable privacy region is the intersection of the two half-planes defined by the standard constraints:

$$1 - \alpha \leq e^\varepsilon \beta \quad \text{and} \quad 1 - \beta \leq e^\varepsilon \alpha. \tag{19}$$

These constraints form the standard piecewise linear trade-off function of $\mathcal{M}_{RR}$.

**2. The RDP Limit:** For the Symmetric Randomized Response mechanism with parameter $p > 0.5$, the exact RDP profile is given by (21). We examine the asymptotic behavior of the constraint boundary as $\tau \to \infty$. Recall the constraint inequality:

$$\alpha^\tau (1 - \beta)^{1-\tau} + (1 - \alpha)^\tau \beta^{1-\tau} \leq \exp((\tau - 1)\rho(\tau)).$$

Substituting $\rho(\tau)$ from (21), the RHS becomes precisely $p^\tau(1 - p)^{1-\tau} + (1 - p)^\tau p^{1-\tau}$. We take the logarithm of

both sides and normalize by $\tau$ to analyze the dominant terms. For the RHS, as $\tau \to \infty$, the term $p^\tau(1 - p)^{1-\tau}$ dominates because $p > 1 - p$:

$$\lim_{\tau \to \infty} \frac{1}{\tau} \ln (\text{RHS})$$

$$= \lim_{\tau \to \infty} \frac{1}{\tau} \ln \left( p^\tau (1-p)^{1-\tau} \left( 1 + \left( \frac{1-p}{p} \right)^{2\tau-1} \right) \right)$$

$$= \ln p - \ln(1 - p) = \ln \frac{p}{1 - p} = \varepsilon.$$

Similarly for the LHS, assuming without loss of generality that $\frac{1-\alpha}{\beta} > \frac{\alpha}{1-\beta}$, the second term dominates:

$$\lim_{\tau \to \infty} \frac{1}{\tau} \ln (\text{LHS}) = \ln(1 - \alpha) - \ln \beta = \ln \frac{1 - \alpha}{\beta}.$$

Thus, the inequality converges to $\ln \frac{1-\alpha}{\beta} \leq \varepsilon \implies 1 - \alpha \leq e^\varepsilon \beta$. Applying the same logic to the symmetric constraint yields $1 - \beta \leq e^\varepsilon \alpha$. These recover exactly the linear boundaries of $R_{\text{DP}}(\varepsilon, 0)$.

**3. The Intersection:** By Lemma 4.2, the attainable region lies in the intersection $\mathcal{R}_{\text{joint}}$. Since the limit $\tau \to \infty$ is included in the family of constraints, $\mathcal{R}_{\text{joint}}$ must be contained within the linear region defined by the limit (Pure DP). Conversely, since $\mathcal{M}_{RR}$ satisfies these RDP constraints for all finite $\tau$, the linear region is contained within every curved RDP region.

Therefore, the intersection $\bigcap_{\tau \geq 0.5} R_{D_\tau}$ exactly recovers the linear region $R_{D_\infty}$. This proves that the conversion rule that utilizes the intersection of RDP privacy regions is tight for RR. □

## 5. Conclusion

In this work, we have established the fundamental limit of black-box conversions from Rényi DP to $f$-DP: we have demonstrated that the conversion rule based on the intersection of privacy regions constitutes the "End of the Road" for RDP-to-$f$-DP conversion research. Our analysis proves that the construction defined by the intersection of single-order regions, equivalent to taking the pointwise maximum across orders in $f$-DP space. Specifically, we have shown that no admissible black-box conversion rule that takes only the RDP profile $\rho(\cdot)$ as input can yield a strictly tighter trade-off function. Consequently, the intersection bound captures all the information contained in the RDP profile, implying that any further improvement would require further information about mechanism parameters.

Furthermore, our analysis reveals the structural simplicity of the "worst-case" mechanisms that saturate this optimal bound. We identified that the boundary is defined by simple Bernoulli processes, with parameters determined by the specific order $\tau$ whose constraint is active at a given query point.

This finding mirrors the folklore in pure Differential Privacy that Randomized Response is the least private mechanism for a fixed budget, effectively extending this intuition to the entire RDP spectrum. This structural insight confirms that the theoretical limit is not an abstract artifact, but a tangible boundary realized by concrete, *elementary* mechanisms.

Practically, this result simplifies the implementation of optimal accounting. To compute the optimal $f$-DP curve from an RDP profile, one does not need solve complex variational problems; it suffices to compute the family of analytic, convex single-order curves and take their pointwise maximum. In practice, this is done over a dense finite grid of orders; we discuss the numerical strategy and its efficiency in Appendix C. We provide a numerically stable code implementation at https://github.com/Felipe-Gomez/Renyi-to-ROC.

As shown by Sommer et al. (2018), mechanisms with countable support can be converted tightly between RDP and $f$-DP. Our conversion is optimal for *any* mechanism described solely by its RDP profile (black box), but it is important to note that it is not guaranteed to be tight for all such mechanisms. For instance, the converted trade-off function for the Gaussian mechanism remains a loose lower bound compared to its analytical form (see Figure 3). We regard discovering mechanism classes for which the black-box conversion is near-optimal as a fruitful direction for future work.

In conclusion, by definitively closing the gap between the upper bounds derived from RDP constraints and the lower bounds realizable by concrete mechanisms, we have resolved the black-box conversion problem.

## Impact Statement

Our work is purely theoretical and advances the understanding of privacy-preserving machine learning. We foresee no specific negative societal consequences as a result of our work.

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

## A. Derivation of the Region Constraints

We now formally derive the constraints (9)–(11). Recall that the decision rule induces two Bernoulli distributions:

$$\mathcal{B}_P \sim \text{Bern}(\alpha) \implies \mathbb{P}_{\mathcal{B}_P}(1) = \alpha, \quad \mathbb{P}_{\mathcal{B}_P}(0) = 1 - \alpha,$$
$$\mathcal{B}_Q \sim \text{Bern}(1 - \beta) \implies \mathbb{P}_{\mathcal{B}_Q}(1) = 1 - \beta, \quad \mathbb{P}_{\mathcal{B}_Q}(0) = \beta.$$

The RDP condition $D_\tau(P\|Q) \leq \rho$ combined with the DPI implies $D_\tau(\mathcal{B}_P\|\mathcal{B}_Q) \leq \rho$. By definition of the Rényi divergence:

$$D_\tau(\mathcal{B}_P\|\mathcal{B}_Q) = \frac{1}{\tau - 1} \log \left( \sum_{z \in \{0,1\}} \mathbb{P}_{\mathcal{B}_P}(z)^\tau \mathbb{P}_{\mathcal{B}_Q}(z)^{1-\tau} \right) \leq \rho. \tag{20}$$

**Case 1: $\tau > 1$**
For $\tau > 1$, the factor $\frac{1}{\tau-1}$ is positive. We can exponentiate both sides without flipping the inequality:

$$\sum_{z \in \{0,1\}} \mathbb{P}_{\mathcal{B}_P}(z)^\tau \mathbb{P}_{\mathcal{B}_Q}(z)^{1-\tau} \leq e^{(\tau-1)\rho}$$
$$\iff \mathbb{P}_{\mathcal{B}_P}(1)^\tau \mathbb{P}_{\mathcal{B}_Q}(1)^{1-\tau} + \mathbb{P}_{\mathcal{B}_P}(0)^\tau \mathbb{P}_{\mathcal{B}_Q}(0)^{1-\tau} \leq e^{(\tau-1)\rho}.$$

Substituting the Bernoulli parameters $\alpha$ and $\beta$:

$$\alpha^\tau (1 - \beta)^{1-\tau} + (1 - \alpha)^\tau \beta^{1-\tau} \leq e^{(\tau-1)\rho}.$$

The second inequality in (9) follows from the symmetric privacy constraint $D_\tau(Q\|P) \leq \rho$, which implies $D_\tau(\mathcal{B}_Q\|\mathcal{B}_P) \leq \rho$. This swaps the roles of the distributions:

$$(1 - \beta)^\tau \alpha^{1-\tau} + \beta^\tau (1 - \alpha)^{1-\tau} \leq e^{(\tau-1)\rho}.$$

**Case 2: $\tau = 1$ (KL-Divergence)**
By definition of the KL-divergence, (20) becomes:

$$\sum_{z \in \{0,1\}} \mathbb{P}_{\mathcal{B}_P}(z) \log \frac{\mathbb{P}_{\mathcal{B}_P}(z)}{\mathbb{P}_{\mathcal{B}_Q}(z)} \leq \rho$$
$$\iff \alpha \log \frac{\alpha}{1 - \beta} + (1 - \alpha) \log \frac{1 - \alpha}{\beta} \leq \rho.$$

Similarly, the symmetric constraint $D_{\text{KL}}(\mathcal{B}_Q\|\mathcal{B}_P) \leq \rho$ yields:

$$(1 - \beta) \log \frac{1 - \beta}{\alpha} + \beta \log \frac{\beta}{1 - \alpha} \leq \rho.$$

**Case 3: $0 < \tau < 1$**
For $\tau < 1$, the term $\frac{1}{\tau-1}$ is negative. Isolating the log-sum reverses the inequality direction:

$$\log \left( \sum_{z \in \{0,1\}} \mathbb{P}_{\mathcal{B}_P}(z)^\tau \mathbb{P}_{\mathcal{B}_Q}(z)^{1-\tau} \right) \geq (\tau - 1)\rho.$$

Exponentiating both sides yields:

$$\sum_{z \in \{0,1\}} \mathbb{P}_{\mathcal{B}_P}(z)^\tau \mathbb{P}_{\mathcal{B}_Q}(z)^{1-\tau} \geq e^{(\tau-1)\rho}$$
$$\iff \mathbb{P}_{\mathcal{B}_P}(1)^\tau \mathbb{P}_{\mathcal{B}_Q}(1)^{1-\tau} + \mathbb{P}_{\mathcal{B}_P}(0)^\tau \mathbb{P}_{\mathcal{B}_Q}(0)^{1-\tau} \geq e^{(\tau-1)\rho}.$$

Substituting the Bernoulli parameters $\alpha$ and $\beta$:

$$\alpha^\tau(1-\beta)^{1-\tau} + (1-\alpha)^\tau\beta^{1-\tau} \geq e^{(\tau-1)\rho}.$$

As in Case 1, the symmetric constraint $D_\tau(\mathcal{B}_Q\|\mathcal{B}_P) \leq \rho$ provides the second inequality required for (11).

Finally, we justify the restriction to $\tau \geq 0.5$. Let $0 < \tau < 0.5$ and $\tau' = 1 - \tau$. Consider the left-hand side of the second inequality in (11) (derived from $D_\tau(Q\|P)$):

$$L(\tau) := (1-\beta)^\tau\alpha^{1-\tau} + \beta^\tau(1-\alpha)^{1-\tau}.$$

Substituting $\tau = 1 - \tau'$, we observe:

$$L(1-\tau') = (1-\beta)^{1-\tau'}\alpha^{\tau'} + \beta^{1-\tau'}(1-\alpha)^{\tau'} = \alpha^{\tau'}(1-\beta)^{1-\tau'} + (1-\alpha)^{\tau'}\beta^{1-\tau'}.$$

This is exactly the left-hand side of the first inequality for $\tau'$ (derived from $D_{\tau'}(P\|Q)$). Thus, the quantity being constrained is identical due to the symmetry of the divergence pair.

Next, we compare the lower bounds. Since $\rho > 0$ and $0 < \tau < 0.5$, we have $\tau - 1 < -\tau = \tau' - 1$. The exponential function is strictly increasing, so:

$$e^{(\tau-1)\rho} < e^{(\tau'-1)\rho}.$$

Recall that for $\tau, \tau' < 1$, the privacy region is defined by lower bounds. Since the bound for $\tau'$ is strictly larger (tighter) than the bound for $\tau$, the condition imposed by $\tau \in (0, 0.5)$ is effectively redundant whenever the constraint for $\tau' \in [0.5, 1)$ is satisfied. Therefore, it suffices to consider $\tau \in [0.5, \infty)$.

## B. Randomized Response Mechanisms

**Definition B.1** (Symmetric Randomized Response). A Symmetric Randomized Response (RR) mechanism $\mathcal{M}_{\mathrm{RR}}$ with retention parameter $p \in [0.5, 1]$ takes a binary input $x \in \{0, 1\}$ and produces a binary output $y \in \{0, 1\}$. The mechanism preserves the input with probability $p$ and flips it with probability $1 - p$:

$$\mathbb{P}(y = x \mid x) = p, \quad \mathbb{P}(y \neq x \mid x) = 1 - p.$$

The probability of reporting the truth is symmetric for both inputs. The resulting mechanism satisfies pure $(\varepsilon, 0)$-DP with:

$$\varepsilon = \ln\left(\frac{p}{1-p}\right).$$

The optimal hypothesis test for this mechanism yields symmetric Type I and Type II errors:

$$\alpha = 1 - p, \quad \beta = 1 - p.$$

The RDP profile of the Symmetric RR is given by

$$\rho(\tau) = \frac{1}{\tau-1} \ln\left(p^\tau(1-p)^{1-\tau} + (1-p)^\tau p^{1-\tau}\right). \tag{21}$$

**Definition B.2** (Asymmetric Randomized Response). The Asymmetric Randomized Response mechanism $\mathcal{M}_{\mathrm{RR}}$ is defined by its transition matrix. For a binary input $x \in \{0, 1\}$ and binary output $y \in \{0, 1\}$, the mechanism operates according to the conditional probabilities $\hat{p}$ and $\hat{q}$, defined as:

$$\begin{aligned}
\mathbb{P}(y = 1 \mid x = 1) &= (1-p)(1-q) =: \hat{p}, \\
\mathbb{P}(y = 1 \mid x = 0) &= p + (1-p)(1-q) =: \hat{q}.
\end{aligned} \tag{22}$$

where $p \in [0, 1]$ is a mixing parameter and $q \in [0, 1]$ is a noise parameter.

The resulting mechanism satisfies pure $(\varepsilon, 0)$-DP, where the privacy budget $\varepsilon$ is determined by the likelihood ratios of these induced probabilities:

$$\varepsilon = \ln\max\left(\frac{\hat{q}}{\hat{p}}, \frac{1-\hat{p}}{1-\hat{q}}\right). \tag{23}$$

Assuming $\hat{q} > \hat{p}$ (which implies the flip dominates), the optimal hypothesis test distinguishing $x = 0$ from $x = 1$ rejects the null when $y = 0$. The resulting error profile is:

$$\alpha = 1 - \hat{q} \quad \text{(Type I)}, \qquad \beta = \hat{p} \quad \text{(Type II)}.$$

*Remark* B.3 (Mechanical Construction via Mixture Model). The probability distribution defined in Definition B.2 is not arbitrary; it arises naturally from a two-stage "mixture" process often used to interpret Randomized Response physically (e.g., in survey methodology (Dwork et al., 2014) or particle processes). Consider the following algorithm:

1. **Mixing Step:** With probability $p$, the mechanism flips the input (mapping $0 \to 1$ and $1 \to 0$) and halts.

2. **Noise Step:** With probability $1 - p$, the mechanism ignores the input entirely and outputs a random bit, reporting $0$ with probability $q$ and $1$ with probability $1 - q$.

This construction provides a structural explanation for the parameters: $p$ controls the correlation with the flipped input (the "deniability" strength), while $q$ controls the bias of the noise.

## C. Numerical Implementation

We provide a numerically stable implementation for computing the optimal $f$-DP curve from an RDP profile, available at https://github.com/Felipe-Gomez/Renyi-to-ROC. To evaluate the optimal $f$-DP bound in practice, we compute the pointwise maximum of single-order trade-off curves $f_{\tau, \rho(\tau)}$ over a dense, finite grid of orders $\tau$. Concretely, we use a logarithmically spaced grid over $\tau \in [0.5, 100]$. The computation is highly efficient: evaluating the trade-off function at one thousand Type I error values $\alpha$ takes on the order of a few milliseconds per order, so sweeping over one hundred orders requires only a few hundred milliseconds in total.

