# OpenReview forum: "Optimal conversion from Rényi Differential Privacy to $f$-Differential Privacy"
_ICML.cc/2026/Conference — ICML 2026 regular_

### Official Review · Reviewer_nAWx · 2026-02-21

**Soundness:** 4
**Presentation:** 4
**Significance:** 3
**Originality:** 2
**Overall Recommendation:** 5
**Confidence:** 4

**Summary:**

The authors study the task of converting differential privacy guarantees from RDP to f-DP. They show that the conversion rule proposed in prior work is optimal both for converting a single RDP guarantee and when converting the full RDP profile. The authors implement a numerically stable converter and provide the source code (link was redacted for the submission).

**Compliance With Llm Reviewing Policy:**

Affirmed.

**Final Justification:**

The authors actively addressed questions and concerns during the rebuttal and discussion phase, and I remain positive about this submission.

The main concern of the other reviewers is the lack of technical novelty. I agree that it is the biggest limitation of the paper. In my opinion, the novelty of this work is insufficient to justify, say, a spotlight or oral presentation at ICML. However, I still think this is a nice contribution. The authors consider a clear research question with theoretical and practical implications and answer it in a concise manner. I see the simplicity of the proof technique as a strength rather than a weakness, because it makes the theoretical contribution more accessible. Additionally, the authors provide a numerically stable implementation which could be a useful tool for practitioners (I have not been able to inspect the code as it was removed for anonymity, but I have no reason to doubt their claims).

The clarity can be improved as pointed out by the other reviewers. But in my opinion there are no major presentation concerns which cannot be addressed in a potential camera-ready version.

**Key Questions For Authors:**

How does the implementation compute the f-DP bound? You state that you can simply compute the single-order curves and take the point-wise maximum. That works well if you have just a few RDP orders, but what happens when you have the full curve e.g. in Figure 3. Do you pick a few RDP orders to get a sufficiently good approximation? A short note in the conclusion or the appendix would be helpful.

Does your work imply optimal conversion for zCDP as well? The standard zCDP definition does not include the extended domain, but otherwise it seems like a common special case of your setting.

In 2.1 you state: "However, we assume that \rho(·) is a valid RDP profile, meaning that the set of mechanisms satisfying the (\tau,\rho(\tau))-RDP constraints is non-empty." I don't understand this assumption. When is \rho(·) not a valid RDP profile? Doesn't any constant function (i.e. perfect privacy) make all \rho(·) valid?

The figures are very nice, but please fix these minor issues: In Figure 1 the legend for asymmetric RR is green, but you use 6 different colors for asymmetric RR. The caption for Figure 3 should include the value of \sigma.

**Limitations:**

The f-DP guarantees obtained with this technique are not necessarily tight for specific mechanisms. The authors discuss this adequately and provide a simple example in Figure 3. This is a general limitation of analyzing algorithms in RDP.

**Strengths And Weaknesses:**

This is a nice result which implies that we can no longer achieve stronger privacy accounting by improving conversion rules from RDP.

The paper is well written and easy to follow. The optimality proof is surprisingly simple. The authors show that any point on the f-DP curve is realized by a randomized response mechanism which satisfy the RDP guarantee. I found it natural that this approach could show optimality for a single curve, but I did not expect that this would be sufficient to show optimality for the full profile.

The only "weakness" is that originality is arguably limited.

---

> ### Author Rebuttal · Authors · 2026-03-30
>
> Thank you for your constructive and thoughtful review and your strong support of our work.
>
> **Questions concerning the implementation:**
> Thank you for this question. As you correctly assumed, we pick a dense finite subset of orders $\tau$. Our implementation yields very fast conversions on a single commodity CPU-core, even for a large number of orders. For example, a trade-off function with one thousand Type I error $\alpha$ values can be computed in a few milliseconds per order, so the evaluation over e.g. one hundred orders log spaced from 0.5 to 100 only takes a few hundred milliseconds. We will add a short paragraph to the Appendix detailing this numerical optimization strategy.
>
> **Question: "Does your work imply optimal conversion for zCDP as well?..."**
>
> Yes, our work implies optimal conversion for zero-Concentrated Differential Privacy (zCDP). A mechanism satisfying $\gamma$-zCDP implies a specific parametric RDP profile $\rho(\tau) = \tau \gamma, \tau \geq 1$. Because our theorem holds for any valid profile $\rho(\cdot)$, substituting the zCDP profile into our intersection bound yields the optimal black-box (i.e., with only the zCDP profile known) $f$-DP curve for zCDP as well.
>
> **Question: "...When is $\rho(\cdot)$ not a valid RDP profile?..."**
>
> Thank you for this observation. Because the perfectly private mechanism has a Rényi divergence of 0 for all orders, any non-negative function $\rho(\cdot)$ trivially upper bounds it, meaning that the set of compliant mechanisms is never empty, and therefore the statement was not strictly required. We will amend this in the final manuscript.
>
> On your question about invalid RDP profiles: When we refer to an RDP profile that *exactly* (i.e., with equality) represents the privacy of some mechanism rather than just a trivial upper bound, $\rho(\cdot)$ is *not a valid profile* if it violates the fundamental analytical properties of Rényi divergence, such as monotonicity. Furthermore, it is invalid if it violates the properties of reparameterized cumulant generating functions, as RDP profiles $\rho(\cdot)$ are reparameterized cumulant generating functions of the privacy loss random variable (see [4]).
> We will include a concise comment to this effect in the revised manuscript and thank you for pointing this out.
>
> **Comment on figure fixes:**
> Thank you for catching these details. We will fix the legend in Figure 1 to correctly map the asymmetric RR colors to the caption. Moreover, in Figure 3, we will also explicitly state that $\sigma=1$.
>
> **References:** [4] Wang, Y. & Balle, B. & Kasiviswanathan, S. (2018) Subsampled Rényi Differential Privacy and Analytical Moments Accountant.

---

> > ### Author Rebuttal · Reviewer_nAWx · 2026-04-03
> >
> > Thank you for the clear responses to my questions.
> >
> > I suggest adding a short note on the implications for zCDP. It's easy for anyone familiar with the definitions to see that zCDP is just a special case of your work. But people unfamiliar with the technical details still use zCDP due to the nice accounting properties e.g. when combining (discrete) Gaussian noise and the exponential mechanism. A short note would broaden the accessibility of your work.

---

### Official Review · Reviewer_BPCR · 2026-03-11

**Soundness:** 3
**Presentation:** 2
**Significance:** 2
**Originality:** 2
**Overall Recommendation:** 3
**Confidence:** 4

**Summary:**

This paper studies conversions from Rényi DP (RDP) to f-DP.
For a fixed pair $(\tau, \rho)$, let $R_{D_\tau}(\rho)$ denote the privacy region defined as the set of all error pairs $(\alpha, \beta)$ attainable by any binary hypothesis test trying to distinguish between any pair $P, Q$ that satisfies $D_\tau(P \| Q) \le \rho(\tau)$ and $D_\tau(Q \| P) \le \rho(\tau)$.
Let $f_{\tau, \rho}$ be the lower boundary of $R_{D_\tau}(\rho)$.
This paper shows $f_{\tau, \rho}(\alpha)$ is optimal trade-off function (in Proposition 3.1).
Then they proceed to show that $\alpha \mapsto \sup_{\tau \ge 0.5} f_{\tau, \rho}(\alpha)$ is the optimal trade-off function for the privacy region of the RDP profile $\tau \mapsto \rho(\tau)$.

**Compliance With Llm Reviewing Policy:**

Affirmed.

**Final Justification:**

The rebuttal addressed my main concerns but reinforced my prior assessment.
The question studied by this paper is meaningful, and the results are interesting, but the technical novelty and non-triviality of this paper make its significance questionable. Also, I think this paper definitely needs to improve the clarity of the presentation of the main results, in particular Proposition 3.1.
Overall, I slightly lean towards rejection.

**Key Questions For Authors:**

- Maybe I'm missing something important. On page 4, you've proved the lower boundary $f_{\tau, \rho}$ of $R_{D_\tau}(\rho)$ is a valid trade-off function. As $f_{\tau, \rho}$ is the boundary, should this give you Proposition 3.1 directly?
- I don't quite understand why do we need to extend the domain of $\tau$ to $\tau < 1$. If we only care about mapping RDP profiles $\tau \mapsto \rho(\tau)$ that are only defined for $\tau \ge 1$, do we still need to consider the privacy region $R_{D_\tau}(\rho)$ for $\tau < 1$ ?

I would be willing to raise my rate, if my confusion/questions are addressed.

**Limitations:**

yes

**Strengths And Weaknesses:**

The problem of converting RDP to f-DP, more generally conversions between different privacy notions, is important.
This paper showed an optimal black-box conversion from RDP to f-DP.
But it seems to me the core argument that underlies all statements is that, among all pair of distributions that satisfies an RDP bound, Bernoulli distributions are the worst. (Other arguments in the paper look quite standard to me.)

---

> ### Author Rebuttal · Authors · 2026-03-30
>
> Thank you for your thoughtful review and your questions. We appreciate the opportunity to clarify your concerns below.
>
>
> **Question 1: "...As $f_{\tau,\rho}$ is the boundary, should this give you Proposition 3.1 directly?":**
> Thank you for this question.
> The fact that the lower boundary $f_{\tau,\rho}$ is a valid trade-off function does not directly yield Proposition 3.1.
>
> Proving that the boundary is "valid" merely establishes that it successfully lower-bounds the trade-off functions of all mechanisms satisfying the $(\tau,\rho)$-RDP guarantee (i.e., $f_{\tau,\rho}$ does not violate the privacy guarantee and includes all mechanisms).
> However, we still require optimality.
> For instance, the trivial function $f(\alpha) = 0$ for $\alpha > 0$ and $f(0)=1$ (representing a complete lack of privacy, i.e., the blatantly non-private mechanism) would also yield a valid conversion (as it is also a valid lower bound).
>
> Therefore, Proposition 3.1 is required to prove that the converted trade-off function is not just valid, but also "optimal", i.e., no tighter valid trade-off function can be constructed solely using the $(\tau,\rho)$-RDP guarantee.
> To prove this optimality, we demonstrate that for every point $(\alpha, \beta)$ on the boundary $f_{\tau,\rho}$, there exist specific (witness) mechanisms, described by Bernoulli pairs of distributions, whose trade-off functions exactly touch the boundary at that specific point.
> Because the trade-off functions of these witness mechanisms saturate the curve (i.e., for each point in the converted trade-off function there exist one mechanism that touches the converted trade-off function at that point), we prove that the theoretical lower bound *cannot be any higher* (above and to the right in trade-off function space) without erroneously excluding valid mechanisms.
>
> We will make this distinction clearer at the beginning of Section 3 and thank you again for this comment.
>
> **Question 2: Necessity of the Extended Domain $\tau \in [0.5, 1)$:**
> Thank you for this important question.
> We need the extended range because evaluating and incorporating the constraints for $\tau \in [0.5, 1)$ yields strictly tighter (better) conversions in certain cases.
>
> In fact, a specific example showing that omitting the full range of $\tau$ values yields a suboptimal conversion for the Gaussian mechanism is shown in Figure 2.
> The figure illustrates how the boundaries constructed from different orders $\tau$ become active (i.e., provide the tightest trade-off function) at different values of the Type I error $\alpha$ according to our optimal conversion.
>
> Specifically, the blue dashed curve, which corresponds to the trade-off function induced by the $\tau=0.5$ privacy region, determines the behavior of the optimal conversion boundary at the fixed point of the converted trade-off function.
> However note in the Figure that the trade-off curves constructed with $\tau \geq 1$ fall *strictly below and to the left* of the curves constructed including $\tau=0.5$ at the fixed point, meaning that they provide a *strictly looser*, i.e., suboptimal bound.
>
> Therefore, to achieve the optimal $f$-DP conversion from an RDP profile, the extended domain is required. We will make this clearer and formalize it in the final manuscript and thank you for your comment.

---

> > ### Author Rebuttal · Reviewer_BPCR · 2026-04-01
> >
> > Thanks. I'm still confused by your response to question 1, $f_{\tau,\rho}$ is not just a valid trade-off function, it is by definition the boundary of the RDP privacy region. So the trivial function you provided does not make sense, as it is not the boundary of the RDP privacy region.

---

> > > ### Author Response · Authors · 2026-04-02
> > >
> > > Thank you, we believe we understand your point now, and we appreciate the opportunity to clarify.
> > >
> > > Please note that there are two sets that, a priori, must be carefully separated:
> > >
> > > 1. **The "2-cut feasible region" $R_{D_\tau}(\rho)$** : the set of $(\alpha,\beta)$ satisfying the divergence inequalities in Eq. (8). This set is obtained as a *necessary* condition via the data processing inequality (DPI). Its lower bound $f_{\tau, \rho}$ is therefore a valid tradeoff function but optimality does not follow.
> > >
> > > 2. **The *exact* $\tau$-RDP class privacy region**, let us call it $R_{D_\tau}(\rho)^{\text{class}}$: the set of $(\alpha,\beta)$ actually attained by *some* mechanism satisfying $(\tau,\rho)$-RDP and their corresponding optimal binary hypothesis test.
> > >
> > > The DPI gives $R_{D_\tau}(\rho)^{\text{class}} \subseteq R_{D_\tau}(\rho)$. But a priori, the inclusion could be strict, since the DPI can be lossy for non-binary mechanisms.
> > >
> > > What Proposition 3.1 actually establishes is the reverse inclusion $R_{D_\tau}(\rho) \subseteq R_{D_\tau}(\rho)^{\text{class}}$: for every $(\alpha,\beta) \in R_{D_\tau}(\rho)$, the Bernoulli pair $P = \text{Bern}(\alpha)$, $Q = \text{Bern}(1-\beta)$ is itself a valid $(\tau,\rho)$-RDP mechanism (since the DPI holds with equality for binary distributions), and the binary hypothesis test attains exactly $(\alpha,\beta)$. Hence $R_{D_\tau}(\rho) = R_{D_\tau}(\rho)^{\text{class}}$: the 2-cut construction $R_{D_\tau}(\rho)$ is tight at the mechanism level. Once this exactness is established, its lower boundary $f_{\tau,\rho}$ is indeed the optimal conversion, as you suggested.
> > >
> > >
> > > We will revise the manuscript to: (a) initially present Eq. (8) as the "2-cut feasible region" obtained via DPI, (b) state Proposition 3.1 as establishing its exactness via Bernoulli witnesses, and (c) note that optimality of the boundary then follows immediately.
> > >
> > > Thank you for your comment, and we remain available for any further clarification.

---

### Official Review · Reviewer_tFaa · 2026-03-14

**Soundness:** 3
**Presentation:** 2
**Significance:** 2
**Originality:** 2
**Overall Recommendation:** 3
**Confidence:** 3

**Summary:**

This paper establishes the fundamental limit of black-box conversions from Rényi Differential Privacy (RDP) to f-DP, proving that the intersection of privacy regions across all RDP orders provides the optimal trade-off functio.

**Compliance With Llm Reviewing Policy:**

Affirmed.

**Key Questions For Authors:**

(1) What is f_{\rho(.)}?

(2) What is the meaning of blackbox in this paper? Does blackbox imply that the adversary does not know the private mechanism?

**Limitations:**

yes

**Strengths And Weaknesses:**

Strengths：  （1） The paper is to answer an open question in Zhu et, al 2022.  And the logic seems reasonable.

Weakness:

(1) Most of the arguments in this paper are quite straightforward from previous works. Although they are mathematically sound, they are probably not significant. More importantly, the conjecture itself is not significant either.

(2) The randomized response mechanisms in the 2-cuts are pure DP but the RDP and f-DP are approximate DP, i.e., their  (epsilon,delta)-converted version are approximate DP; but the authors used them to show the optimality of the conversion rule. It seems that there is a big conceptual gap in the proof.

(3) Lemma 4.2 is straightforward; I don’t see why the authors include the details of the proof in the paper.

 (4) What is the induced trade-off function f_{\rho(.)}? Induced by what?

 (5) Several figures in the paper are not informative， especially Figure 1 looks confusing to me.

Minor:  The notion of Blackwell equivance is not used after its definition.

---

> ### Author Rebuttal · Authors · 2026-03-30
>
> Thank you for your thorough review. We address your concerns below.
>
> **Significance of the Conjecture:**
> Establishing the fundamental conversion limit of RDP to $f$-DP is a significant theoretical contribution, as echoed by the other reviewers. Definitively closing this open problem provides substantial value to the community for the following reasons:
> * RDP is currently the best, and often the only, tractable accounting tool we have for specific advanced algorithms. For instance, the tightest individual privacy accounting techniques [1] and hyperparameter transfer approaches [2] rely entirely on RDP.
> * For private selection, the exponential mechanism is frequently analyzed via its tight characterization with zero-Concentrated Differential Privacy [3], which maps directly to an RDP profile $\rho(\cdot)$.
>
> However, the mathematical bridge between the analytical accounting of RDP and the exact operational interpretation of $f$-DP has so far been missing. Our paper provides this exact bridge in a provably optimal way. We agree that the significance of our findings could be more clearly described, and we will highlight these applications in the introduction of the camera-ready version.
>
> **Weakness 2:**
> You are correct that the randomized response (RR) mechanisms in the 2-cuts are pure DP, and that a $(\tau, \rho)$-RDP bound is characterized by approximate DP rather than pure DP. The converted trade-off function needs to be a valid *lower bound* (i.e., below and to the left) of all the RR (witness mechanisms) trade-off functions. In Figure 1, the blue trade-off curve is the conversion and the colored lines correspond to the witness mechanisms. The witness mechanisms lie above and to the right of the converted trade-off function and thus do not imply that the converted mechanism satisfies pure DP, otherwise they would need to lie *below and to the left* of the converted trade-off function. This is exactly what makes 2-cuts a powerful theoretical tool: they allow us to analyze a RDP mechanism using RR, despite the mechanism not satisfying pure-DP. Please let us know if we can clarify further.
>
> **Lemma 4.2 (Weakness 3), Weakness 4 and definition of $f_{\rho(\cdot)}$ (Q1):**
> Thank you for this feedback. To streamline the presentation, we will condense Lemma 4.2 into a corollary of Lemma 4.1. Furthermore, we will use this result to explicitly define the trade-off function $f_{\rho(\cdot)}(\cdot)$ as the lower boundary of the joint intersection region $R_\text{joint}$. Formally, this is expressed as $f_{\rho(\cdot)}(\alpha) := \sup_{\tau \geq 0.5} f_{\tau,\rho(\tau)}(\alpha)$.
>
> Regarding the terminology, when we state that a trade-off function is "induced", we mean that the geometric construction of the RDP privacy region naturally forms (or induces) a corresponding trade-off function, which is exactly its lower boundary. Therefore, $f_{\rho(\cdot)}$ is induced by the intersection of all single-order privacy regions defined by the profile $\rho(\cdot)$. We will explicitly clarify the use of the term "induced" in the camera-ready version.
>
> **Clarity of Figures (Weakness 5):**
> Thank you for this feedback. Figure 1 visually demonstrates the core technical innovation underpinning our optimality argument. Specifically, if any proposed conversion rule were to yield a tighter bound, i.e., a trade-off function above and to the right of the bold blue curve, it would incorrectly intersect and thus invalidate the RR witness mechanisms.
> However, because we established via Proposition 3.1 that any universally valid converted trade-off function **must** strictly lie below and to the left of **all** valid RR mechanisms, this would lead to a direct contradiction.
>
> We will add this geometric intuition immediately following Proposition 3.1 in the camera-ready version.
>
> **Meaning of black-box (Q2):**
> In our work, black-box means that the conversion is oblivious to *all* knowledge about the DP mechanism, except for the sole fact that it satisfies RDP for a given profile $\rho(\cdot)$. Thus, the conversion is not allowed to leverage *any mechanism-specific information* (e.g., the knowledge that the underlying mechanism is Gaussian).
> This is formally defined at the beginning of Subsection 2.3. We agree this is a crucial point and are happy to clarify this earlier in the introduction of the camera-ready version.
>
> **Minor comment on Blackwell:**
> Thank you for pointing this out. We will add a more substantial exposition on the exact meaning and importance of Blackwell optimality to the final manuscript as it explains mechanism orderings and equivalences in the most mathematically rigorous way.
>
> **References:**
> [1] Feldman, V., & Zrnic, T. (2021). Individual privacy accounting via a Rényi filter.
> [2] Papernot, N., & Steinke, T. (2022). Hyperparameter Tuning with Rényi Differential Privacy.
> [3] Cesar, M., & Rogers, R. (2021). Bounding, concentrating, and truncating: Unifying privacy loss composition for data analytics.

---

> > ### Author Rebuttal · Reviewer_tFaa · 2026-04-08
> >
> > I am not satisfied with the rebuttal about my second question, i.e, the confusing between pure and approxiamte DP.   From Fig 1, the authors really meant RDP to be pure DP.    In the literature about simulation lemma of DP,  people use leaky RR rather than pure RR.
> >
> > Blackwell should be important to show the optimality. The authors stated it but did not use it in the proofs.

---

> > > ### Author Response · Authors · 2026-04-08
> > >
> > > We appreciate the reviewer's follow-up, but we must respectfully clarify that there are no factual errors or contradictions in our mathematics. The concerns raised stem from fundamentally misapplying the logic of the standard simulation lemma to our distinct bounding approach.
> > >
> > > **1. The precise use of Blackwell Equivalence:**
> > > There appears to be a misunderstanding about how we apply Blackwell's theorem in our work. As stated at the end of Section 2.2, we use the *ordering property* of Blackwell to compare the information content of mechanisms [5]. The reviewer seems to be referring to the use of Blackwell in a post-processing context, specifically to derive the simulation lemma [6]. While both concepts are consequences of Blackwell’s theorem, we do not use the simulation lemma to prove anything. This lemma is simply not used in our work, nor is it necessary for our optimality proof.
> > > Furthermore, our foundational analysis operates natively on **RDP privacy regions** (see Section 2.4.1), whereas the simulation lemma from Kairouz et al. [6] operates on privacy regions induced by the **hockey-stick divergence** (which we also discuss in Section 2.4 of our work).
> > >
> > >
> > >
> > > **2. Proof logic: Upper Envelopes vs. Universal Lower Bounds:**
> > > Because we do not rely on the simulation lemma (where one mechanism can be post-processed into another) or on constructions mapping approximate DP to $f$-DP (such as those by Dong et al. [7]), our proof logic operates fundamentally differently:
> > > * **Converting $(\epsilon, \delta(\epsilon))$-DP to $f$-DP:** To find the exact $f$-DP trade-off for a *single* mechanism, one takes the *upper envelope* of its infinitely many $(\epsilon, \delta)$-DP guarantees (which relies on leaky RR).
> > > * **Our Work:** We derive a universal conversion rule. Therefore, our converted trade-off must serve as a universal *lower bound* for the trade-off functions of *all possible* mechanisms that satisfy a given RDP guarantee.
> > >
> > > **3. Figure 1 does not imply RDP is pure DP:**
> > > Because our goal is to establish a lower bound, Figure 1 does **not** state or imply that RDP is pure DP. The geometry strictly proves the opposite: the colored lines (pure DP 2-cuts) lie *above* the blue line (our RDP bound). For our proof to imply that the underlying RDP mechanism is pure DP, the bounding relationship would have to be reversed (i.e., the colored lines would need to lie *below* the blue line). We merely use the pure DP mechanisms as upper bounding witnesses to prove our universal lower bound cannot be pushed any higher.
> > >
> > > In summary, our work nowhere contradicts the fact that RDP characterizes approximate DP, nor does it contradict prior literature on the simulation lemma. Our mathematical proof is fully sound as written.
> > >
> > >
> > >
> > > **References:**
> > >
> > > [5] Kaissis, G., Kolek, S., Balle, B., Hayes, J., and Rueckert, D. Beyond the calibration point: Mechanism comparison in differential privacy, 2025. URL https://arxiv.org/abs/2406.08918.
> > > [6] Kairouz, P., Oh, S., and Viswanath, P. The composition theorem for differential privacy. In *International Conference on Machine Learning (ICML)*, pp. 1376–1385, 2015.
> > > [7] Dong, J., Roth, A., and Su, W. J. Gaussian differential privacy. *Journal of the Royal Statistical Society: Series B (Statistical Methodology)*, 84(1):3–37, 2022.

---

### Decision · Program_Chairs · 2026-04-30

**Decision:**

Accept (regular)

**Comment:**

This paper studies the problem of converting Rényi Differential Privacy (RDP) guarantees into f-DP guarantees, and proves that the intersection of single-order RDP privacy regions yields an optimal black-box conversion. Reviewers agreed that the question is well-defined and that the result cleanly resolves an open conjecture, providing a unified and conceptually simple characterization of the optimal conversion rule.

The main concern raised by reviewers is that the technical novelty is somewhat limited, as several components of the argument build on prior work and the proof itself is relatively simple. There were also some questions regarding presentation clarity and conceptual framing, which were partly addressed in the rebuttal.

In my view, the paper provides a clear and useful resolution of an important question in privacy accounting. The result establishes an interesting fundamental limitation -- that no stronger black-box conversion from RDP to f-DP is possible, which is both theoretically clean and practically relevant for the design and analysis of private algorithms.

Given that the paper is technically sound, clearly written, and addresses a useful problem for the community, I recommend acceptance.